# Square Planar Pt(II) Ion as Electron Donor in Pnictogen Bonding Interactions

**Sergi Burguera** , **Rosa M. Gomila, Antonio Bauzá** and **Antonio Frontera ***

Department of Chemistry, Universitat de les Illes Balears, Crta. de Valldemossa km 7.5, 07122 Palma, Spain
* Correspondence: toni.frontera@uib.es

**Abstract:** It has been proposed that late transition metals with low coordination numbers (square planar or linear) can act as nucleophiles and participate in σ-hole interactions as electron donors. This is due to the existence, in this type of metal complexes, of a pair of electrons located at high energy d-orbitals ($d_z^2$ or $d_{x^2-y^2}$), which are adequate for interacting with antibonding σ-orbitals [σ*(X–Y)] where Y is usually an electron withdrawing element and X an element of the p-block. This type of d[M]→σ*(X–Y) interaction has been reported for metals of groups 9–11 in oxidation states +1 and +2 ($d^8$ and $d^{10}$) as electron donors and σ-holes located in halogen and chalcogen atoms as electron acceptors. To our knowledge, it has not been described for σ-holes located in pnictogen atoms. In this manuscript, evidence for the existence of pnictogen bonding involving the square planar Pt(II) metal as the electron donor and Sb as the electron acceptor is provided by using an X-ray structure retrieved from the Cambridge Structural Database (CSD) and theoretical calculations. In particular, the quantum theory of atoms in molecules (QTAIM), the noncovalent interaction plot (NCIPlot) and molecular electrostatic potential (MEP) methods were used. Moreover, to further confirm the nature of the Sb···Pt(II) contact, a recently developed method was used where the electron density (ED) and electrostatic potential (ESP) distribution were compared along the Sb···Pt(II) bond path.

**Keywords:** σ-hole interactions; pnictogen bonding; crystal engineering; platinum; X-ray structures; DFT calculations; QTAIM analysis

## 1. Introduction

Noncovalent interactions are very important in many fields, including chemistry, biochemistry and supramolecular materials. In particular, the utilization of σ-hole interactions [1] as directing forces in several areas of modern chemistry, such as crystal engineering [2], supramolecular chemistry [3] and supramolecular catalysis [4–6], is constantly growing. The most used and studied interaction is the halogen bonding (HaB), where any element of group-17 plays the role of the electrophile (electron acceptor, σ-hole donor) [7]. The HaB is ideal to be used in crystal engineering [8] due its strong directionality, especially between iododerivatives and conventional lone pair (LP) donor atoms such as N and O. Moreover, chalcogen bonding interactions (ChB, group 16 element acting as σ-hole donor), recently defined by the International Union of Pure and Applied Chemistry (IUPAC) [9], has experienced a fast growth in the last five years as a convenient supramolecular force to be used in supramolecular catalysis, crystal engineering and anion recognition [10–14]. Other σ-hole interactions such as pnictogen bonding [15–17] and tetrel bonding [17–19] are less developed, though several works have evidenced their potential use in supramolecular catalysis and as directing forces in crystal engineering [6,15–19].

In the last two years, several works and reviews have evidenced that late transition metals (groups 9–11) with low coordination numbers (linear and square planar) can also participate in σ-hole and π-hole interactions as electron donors [20–25]. This is based on the fact that $d^8$ and $d^{10}$ transition metals have high energy lone pairs located in $d_z^2$ and

$d_x{}^2\text{-}_y{}^2$ orbitals that provide enough nucleophilicity to the metal center. These counterintuitive d[M]→σ*(X–Y) interactions have been reported for halogen [21,26] and chalcogen bonds [27] but have not been described before for pnictogen bonding, as far as our knowledge extends (see Scheme 1). The covalent (coordination bond) version of the Pt–Sb bond has been analyzed before and divided into three different bonding modes: purely covalent (also known as ligand X-bond), Pt→Sb dative bond (also known as ligand Z-bond) and Pt←Sb dative bond (also known as ligand L-bond) [28]. Gabbaï also analyzed coordination bonds involving Sb and late transition metals, introducing the term σ-donor/acceptor-confused ligands [29].

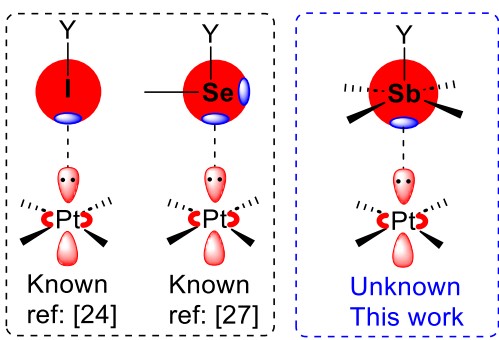

**Scheme 1.** Known halogen and chalcogen bonding (**left**) and unknown pnictogen bonding (**right**, this work) involving Pt(II) as electron donor.

In this manuscript, several examples of X-ray structures exhibiting Pt···Halogen and Pt···Chalcogen are described to provide a preliminary study on the importance of σ-hole interactions involving metals as donors. Moreover, the existence on PnBs involving five-coordinated Sb and Bi-atoms in different oxidation states (+3 and +5) is emphasized by describing several X-ray structures retrieved from the Cambridge Structural Database (CSD) where Sb/Bi···N and Sb···O PnBs play a relevant role governing their solid-state architecture. More importantly and for the first time in the literature, the existence of Pt(II)···Sb PnBs in an X-ray structure was demonstrated by using several computational tools, which showed a dominant charge transfer from the Pt(II) to the Sb(III), as described in the following sections. Finally, for another structure retrieved from the CSD and also exhibiting a short Pt(II)···Sb contact, the existence of a PnB was ruled out since the most significant charge transfer occurred from the Sb(III) to the Pt(II) atoms, as evidenced by the theoretical analysis.

## 2. Results and Discussion

### 2.1. Preliminary Work on Pt···Halogen and Pt···Chalcogen Bonds

It has been demonstrated theoretically that the Pt(II) metal center is more nucleophillic than Ni(II) and Pd(II) and it is able to participate in halogen bonding with good halogen bond donors such as pentafluoroiodobenzene with binding energies up to 6 kcal/mol [30]. Experimentally, the Pt(II)···I interaction has been used in crystal engineering as shown in Figure 1. In particular, the ROZZUE structure [31] is a 1:1 co-crystal composed by one [Pt(acac)$_2$] (Hacac = acetylacetone) molecule and one 1,3,5-trifluoro-2,4,6-triiodobenzene. In the solid state, the Pt(II) atom establishes two symmetrically equivalent HaBs, where two C–I bonds from two different σ-hole donor molecules point to the Pt atom above and below the molecular plane. The C–I···Pt angle was 168.7°, thus confirming that the σ-hole at the I-atom was indeed interacting with the Pt-atom (HaB nature of the I···Pt contact) instead of the negative belt of iodine. This fact obviously discards the possibility of the formation of a Pt···I coordination bond. Moreover, the long Pt···I distance observed in the solid state (3.45 Å) is typical of a noncovalent distance. The second structure highlighted in Figure 1b also exhibits structure directing C–I···Pt interactions (refcode XUVXOE) [20]. In this case, the σ-hole donor molecule was the ditopic 1,4-diiodotetrafluorobenzene that connected two

dinuclear Pt-complexes. This "half-lantern" complex exhibited an enhanced nucleophilicity of the outer lobes of the $d_z{}^2$ orbitals due to the short Pt⋯Pt distance (2.85 Å). This effect facilitates the formation of supramolecular assemblies governed by strong and directional C–I⋯Pt HaBs (see chemical/orbital diagram in Figure 1). The authors also demonstrated that such contacts occur in the solution by using $^{195}$Pt NMR and UV/Vis studies [20]. The comparison of the assemblies generated in the mononuclear (ROZZUE) and dinuclear Pt(II) (XUVXOE) complexes revealed that the Pt⋯I distance was much shorter in the latter and the Pt⋯I–C angle (174.1°) was closer to linearity, as expected for HaBs. Therefore, the polarization of the $d_z{}^2$ orbitals towards the outer part enhanced the HaBs in XUVXOE in comparison with the mononuclear ROZZUE assembly.

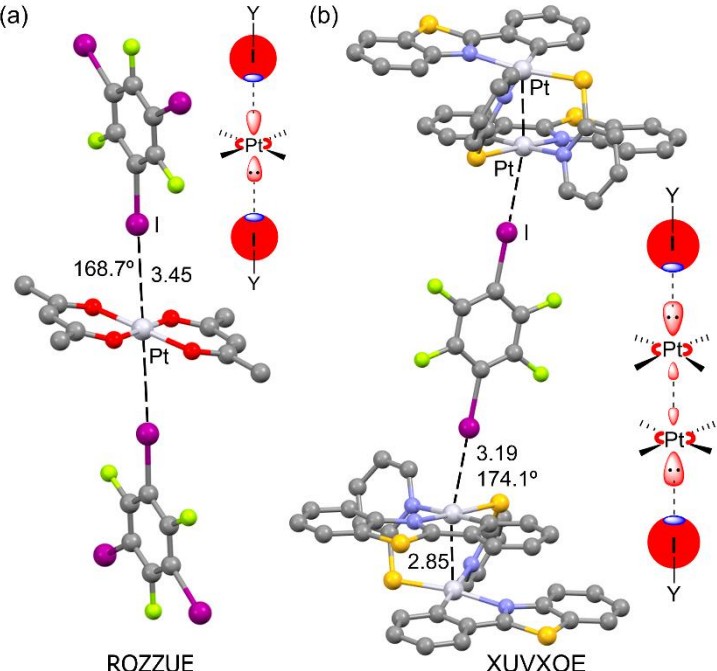

**Figure 1.** X-ray structures of ROZZUE (**a**) and XUVXOE (**b**). H-atoms omitted. Distances are given in Å. Color code: Carbon grey, fluorine light green, platinum light grey, sulfur yellow, iodine purple.

A similar assembly governed by the formation of Pt(II)⋯Se chalcogen bonds was also reported [27] using 4,4′-selanyl*bis*(tetrafluoropyridine) as σ-hole donor and acetylacetonato-(2-(pyridin-2-yl)phenyl)-platinum(II) as cocrystal coformer (see Figure 2a (refcode VALVUD)). Two square planar Pt(II) complexes π-stacked with a Pt⋯Pt distance of 3.30 Å. The π-stacked dimer interacted with two chalcogen bond donor molecules forming a remarkable C–Se⋯Pt/Pt⋯Pt/Pt⋯Se–C quaternary assembly. The Se⋯Pt distance was 3.35 Å and the C–Se⋯Pt angle was 164.3°, which are in the range of typical of noncovalent ChBs [12,14]. Two additional examples of X-ray structures exhibiting Ch⋯Pt(II) interactions are represented in Figure 2. One corresponds to dichloro-(1-methyl-3-[(phenylselanyl)methyl]-imidazol-2-ylidene)-platinum(II) (refcode QETFED) [32] that forms self-assembled dimers in the solid state with two symmetrically equivalent Se⋯Pt interactions (distance 3.69 Å and angle 173.6°). The longer distance observed in this X-ray structure with respect to VALVUD was likely related to the nature of the aromatic ring bonded to the Se-atoms. Whilst in VALVUD, the ring was perfluorinated (electron withdrawing group), and in QETFED, the selenium was bonded to a phenyl ring, thus the Se–C bond was less polarized and the σ-hole opposite to this bond was less intense. Figure 2c shows a structure with tellurium instead of selenium (refcode CUHMAV) [33] that also formed self-assembled dimers in the solid state through the formation of two symmetric Te⋯Pt(II) ChBs with a distance of 3.71 Å and angle of 166.5°.

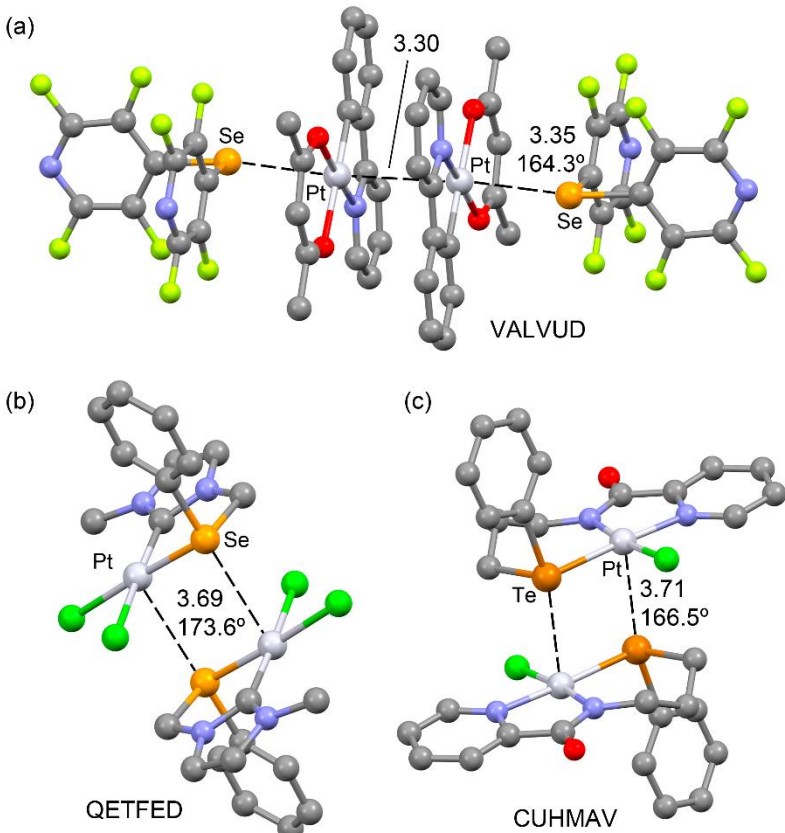

**Figure 2.** X-ray structures of VALVUD (**a**), QETFED (**b**) and CUHMAV (**c**). H-atoms omitted. Distances are given in Å. Color code: Carbon grey, chlorine green, platinum light grey, selenium orange, tellurium tangerine, oxygen red, nitrogen blue.

Several works have reported conventional chalcogen bonds in Pd(II) and Pt(II) complexes focusing either on the enhancement of the σ-hole intensity at the Se/Te-atoms upon coordination to the transition metals [34] or in catalysis [35].

## 2.2. X-ray Structures Exhibiting Sb,Bi···N Pnictogen Bonds

Most theoretical and computational investigations on pnictogen bonding (PnB) involve trivalent pnictogen atoms [15]. However, pentavalent pnictogen atoms in square pyramidal coordination also form directional pnictogen bonds.

To illustrate the ability of pentacoordinated Sb,Bi-atoms in oxidation state +3 to participate in pnictogen bonding, some cocrystal salts with N-atom as donor were gathered, as seen in Figure 3, which included different types of ammonium salts. For instance, AQAMIR [36] is a salt composed by the 1,1′-(pyridazine-3,6-diyl)bis(1H-imidazol-3-ium) dication and pentachloro-antimony(III) dianion. It can be observed that the centroid (Cg) of the N–N bond of pyridazine ring was located opposite to the axial Sb–Cl bond with a distance 3.035 Å and Cg(NN)···Sb–Cl angle close to linearity (172.8°). Both Sb···N distances were 3.11 Å and 3.09 Å, which were significantly shorter than the sum of van der Waals radii ($\Sigma R_{vdw}$ = 3.80 Å) and longer than the sum of covalent radii (2.10 Å), thus confirming the noncovalent nature of the contacts. The DEKNOX [37] structure corresponded to the salt where the cationic part was bis((m-phenol)-1,2,4-triazolium) and the counterion was bis(μ²-iodo)-hexaiodo-di-bismuth $[Bi_2I_8]^{2-}$ dianion. The Bi-atoms of the anion established two symmetrically equivalent Bi···N contacts involving one of the N-atoms of the triazolium ring. The Bi···N distance was quite short ($\Sigma R_{vdw}$ = 3.90 Å), likely due to the electrostatic attraction between the counterions. The Bi···N PnBs were quite directional (172.3°) as expected for this type of σ-hole interaction. Finally, the DOQFIB [38] fragment shown in Figure 3c was composed of 1-methylpyrazinium cation and the counterion was bis(μ-

bromo)-heptabromo-di-bismuth $[Bi_2Br_9]^{3-}$. The non-methylated N-atom of the pyrazinium ring was located opposite to the axial Bi–Br bond with the lone pair pointing to the Bi-atom. The distance and angle (2.840 Å and 172.9°) were similar to those observed in DEKNOX.

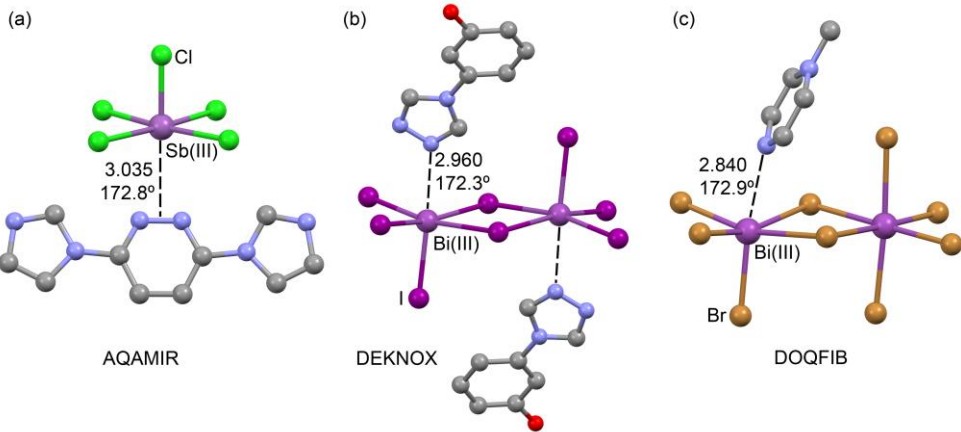

**Figure 3.** X-ray structures of AQAMIR (**a**), DEKNOX (**b**) and DOQFIB (**c**). H-atoms omitted. Distances are given in Å. Color code: Carbon grey, chlorine green, antimony and bismuth violet, bromine bronze, oxygen red, nitrogen blue, iodine purple.

In addition to the co-crystal salts commented above, PnBs involving pentacoordinated Sb(V)/Bi(V) and Sb(III) are represented in Figure 4. They corresponded to either solvates involving Sb(V)/Bi(V) derivatives or a discrete Sb(III) molecule where the PnBs were formed between neutral components. For example, CEKPAK [39] corresponded to (3,6-di-t-butyl-4,5-dimethoxycatecholato)-triphenyl-antimony(V) acetonitrile solvate, where the N-atom of the solvent molecule points to the Sb(V) atom establishing a very short (2.775 Å) and directional ($\angle$ N···Sb–C = 177.0°) PnB. Interestingly, these Sb(V) derivatives are able to reversibly bind molecular oxygen to yield spiroendoperoxide due to the electrophilicity of the Sb(V) atom.

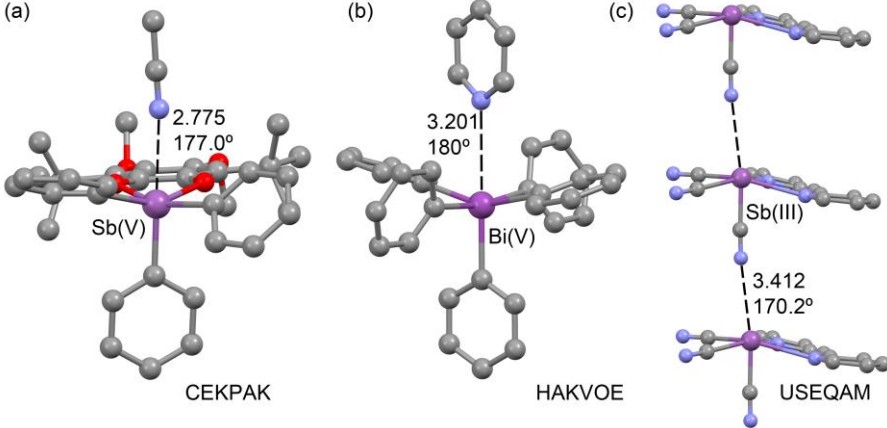

**Figure 4.** X-ray structures of CEKPAK (**a**), HAKVOE (**b**) and USEQAM (**c**). H-atoms omitted. Distances are given in Å. Color code: Carbon grey, platinum light grey, antimony and bismuth violet, oxygen red, nitrogen blue.

Figure 4b shows the X-ray structure of pentaphenyl-bismuth pyridine solvate [40] (refcode HAKVOE), where the N-atom of pyridine pointed exactly opposite to the Bi–C bond at a typical noncovalent distance (3.201 Å). Finally, it is worth highlighting the USEQAM [41] structure (see Figure 4c), where highly directional PnBs (Sb···N = 3.412 Å and $\angle$ N···Sb–C = 170.2°) PnBs propagated the (2,2′-bipyridine)-tricyano-antimony(III) molecules to 1D supramolecular polymers. In particular, the N-atom of one of the cyanide

ligands interacted with the Sb(III) atom of the adjacent molecule, thus propagating the polymer. The starting $Sb(CN)_3$ product used to synthesize the USEQAM compound had three deep σ-holes [2,42] opposite to the Sb–CN bonds. Two of them were used to interact with the 2,2′-bipyridine ligand forming coordination (mostly covalent) Sb–N bonds and the third one was used to establishing the noncovalent PnB contact.

### 2.3. X-ray Structures Exhibiting Sb···O Pnictogen Bonds

Three examples are shown in Figure 5 for the oxygen atom acting as Lewis Base, where the Sb-atom was either in oxidation state +5 (refcode CEKNUC) or +3 (refcodes HEKQUN and RAHFEN). The CEKNUK [39] structure is a methanol solvate of (3,6-di-t-butyl-4-methoxycatecholato)-triphenyl-antimony(V), similar to the acetonitrile solvate shown in Figure 5, where the O-atom of methanol was precisely located opposite to the axial Sb–C bond. The Sb···O distance was very short due to the strong acidity of the Sb(V) atom. In case of HEKQUN [43] and RAHFEN [44] structures, which are N,N-dimethylformamide (DMF) and 1,4-dioxane solvates, respectively, both the $[SbCl_5]^{2-}$ and the $[Sb_2I_8]^{2-}$ moieties are dianionic and consequently the PnBs are rather counterintuitive. That is, the anions are acting as Lewis acids, as can be observed in Figure 5, establishing rather short Sb···O PnBs with the solvent molecules that were the Lewis bases. In particular, the O-atom of the DMF was located opposite to the apical Sb–Cl bond of the $[SbCl_5]^{2-}$ anion (Sb···O distance = 3.27 Å and Cl–Sb···O angle = 166.3°) in the HEKQUN structure (see Figure 4b). Similarly, the $[Sb_2I_8]^{2-}$ anion established two symmetrically equivalent Sb···O PnBs with the 1,4-dioxane molecules in the RAHFEN structure, where the equatorial oxygen lone pairs interacted with the apical Sb–I bonds.

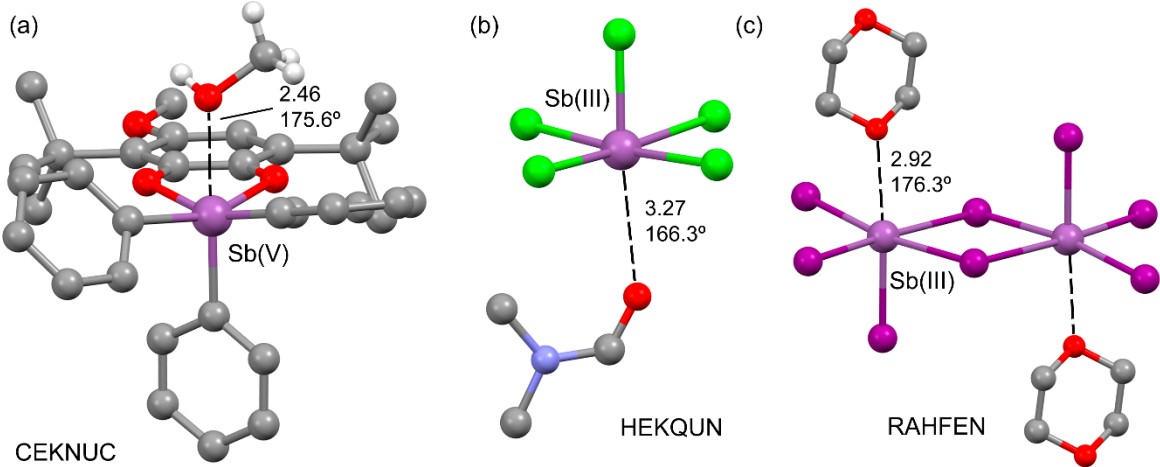

**Figure 5.** X-ray structures of CEKNUC (**a**), HEKNUC (**b**) and RAHFEN (**c**). H-atoms omitted apart from the methanol molecule in CEKNUC. Distances are given in Å. Color code: Carbon grey, chlorine green, antimony violet, oxygen red, nitrogen blue, iodine purple.

### 2.4. X-ray Structures Exhibiting Sb···Pt Pnictogen Bonds

Figure 6a shows the X-ray structure of RUVBIU [45], which was a salt formed by $[SbCl_5]^{2-}$ dianion and square-planar $[Pt(L)(TTCN)]^{2+}$ dication where L was 5-nitro-1,10-phenanthroline and TTCN was 1,4,7-trithiacyclononane. It can be observed that the Pt(II) metal center of the cation was located approximately opposite to the apical Sb–Cl bond of the $[SbCl_5]^{2-}$ unit, thus establishing a Sb···Pt(II) contact. Moreover, the non-coordinated S-atom of TTCN was located at the opposite side of the $PtN_2S_2$ plane, also interacting with the Pt(II), likely polarizing the $d_z{}^2$ orbital and increasing its nucleophilicity. The MEP surface of the dianion is represented in Figure 6b, evidencing the presence of a negative σ-hole opposite to the Sb–Cl bond (the MEP surface of the cationic part is given in the supplementary file, Figure S1). Taking into consideration the strong electrostatic attraction between the doubly charged counterions, the Pt(II)···Sb interaction can be defined as a

"charge unassisted" PnB since for this particular contact the anion was acting as the electron acceptor and the cation was acting as electron donor. The combination of QTAIM and NCIplot analyses is given in Figure 4c, showing that the anion and cation are interconnected by several bond critical points (CPs, red spheres) and bond paths, thus evidencing the contribution of several contacts. Remarkably, a bond CP and bond path connects the Pt(II) atom to the Sb atom, thus confirming the existence of an interaction between both atoms. Moreover, a blue disk-shape NCIplot isosurface is also located between both atoms, coincident to the position of the bond CP. The color of the reduced density gradient (RDG) isosurface of the Pt···Sb contact (blue is used herein for strongly attractive interactions) compared to the colors of the RDG isosurfaces that characterize the rest of contacts (green is used herein for weakly attractive interactions) clearly suggests that the Pt···Sb was the strongest interaction and likely dictated the final geometry adopted by the dimer. The binding energy was very large (−265.7 kcal/mol) and clearly dominated by the electrostatic attraction between the counterions (pure and non-directional coulombic attraction). The dispersion contribution to this binding energy is very important, since the interaction energy was significantly reduced to −219.5 kcal/mol if the dispersion correction was not considered. Moreover, if relativistic effects were included in the basis set (see theoretical methods below), the interaction energy increased to −281.5 kcal/mol, thus revealing that such effects are important to be considered in systems with heavy atoms such as Pt and Sb, although that is not the topic of the present investigation.

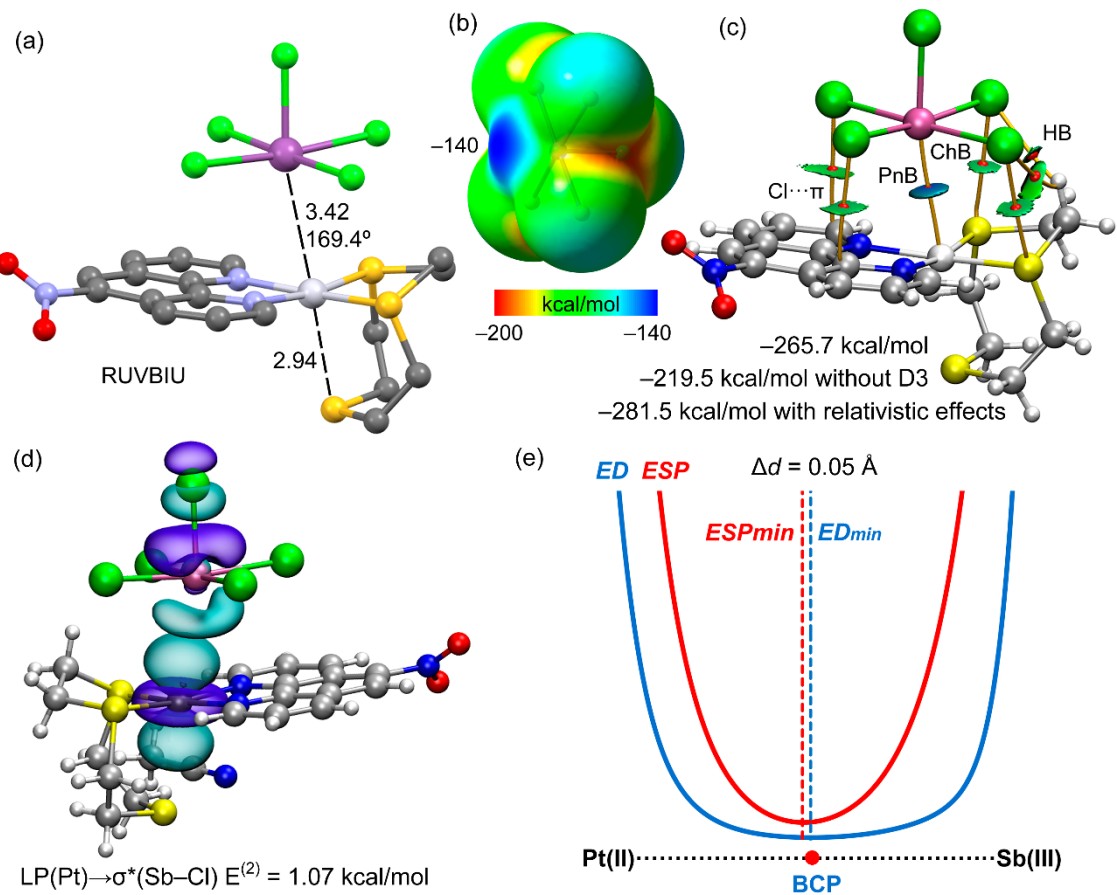

**Figure 6.** (**a**) X-ray structure of RUVBIU. Distances in Å. H-atoms omitted. Color code: Carbon grey, chlorine green, platinum light grey, antimony violet, sulfur yellow, oxygen red, nitrogen blue, iodine purple. (**b**) MEP surface of the $[SbCl_5]^{2-}$ unit. (**c**) QTAIM (bond CPs in red and bond paths as orange lines) and RDG isosurfaces of the salt. (**d**) NBO representation of the LP(Pt)→σ*(Sb–Cl) interaction. (**e**) ED/ESP profile along the Pt(II)···Sb(III) bond path in RUVBIU.

In an effort to demonstrate the "reverse" charge transfer from the Pt(II) ion to the $[SbCl_5]^{2-}$ dianion, the natural bond orbital (NBO) analysis was carried out. The result of the NBO analysis is indicated in Figure 4d, evidencing a small orbital contribution ($E^{(2)}$ = 1.07 kcal/mol) due to a donation from an LP orbital of Pt(II) located at the $d_z{}^2$ atomic orbital to the antibonding Sb–Cl orbital σ*(Sb–Cl), as is common in σ-hole interactions, and demonstrating the charge transfer from the Pt(II) to the Sb. In addition, there was an additional orbital interaction from the bonding Pt–N σ orbitals to the antibonding Sb–Cl orbital, σ(Pt–N)→σ*(Sb–Cl) with a concomitant energy of $E^{(2)}$ = 1.15 kcal/mol (see ESI for a complete list of contributions).

To further demonstrate the nucleophilicity of the Pt(II) atom in the RUVBIU structure, a very convenient methodology was used (see Figure 6e). It was based on the comparison of electron density (ED) and electrostatic potential (ESP) distribution along the Pt(II)⋯Sb(III) bond path [46]. In any donor–acceptor interaction, the ED minimum is closer to the electron acceptor atom and the ESP minimum is closer to the electron donor atom [47]. The plot shown in Figure 4e and the positions of ED and ESP minima along the Pt(II)⋯Sb(III) bond path confirm the electron donation of the Pt(II) towards the Sb, thus revealing the PnB nature of the contact and that the Pt(II) is acting as the nucleophile. The distance between both minima are indicated in Figure 6e, which was 0.05 Å.

Another X-ray example was selected with a Pt⋯Sb contact that is represented in Figure 7a (refcode PEPTAS) [48]. It was also a salt where the cationic part was formed by the Pt(II) metal center coordinated to two different diamines (propylene-1,2-diamine and *N,N′*-dimethylethylenediamine). The dianion was constituted by two tartrato tetra-anions bonded to two Sb(III) ions. It can be observed that in the solid state, the Sb was located over the Pt(II) at a distance (3.77 Å) that was much longer than that observed in RUVBIU but still shorter than the sum of the van der Waals radii (4.25 Å) [49]. The largest O–Sb⋯Pt(II) angle was 152.3°, far from linearity, thus suggesting that the σ-hole nature of the contact was not clear. The MEP surface plot represented in Figure 7b evidences that the MEP maximum was indeed located at the Sb-atom (−111 kcal/mol) and the combined QTAIM/NCIplot analysis confirmed the existence of the Pt⋯Sb contact showing the corresponding bond CP, bond path and green RDG isosurface connecting both atoms (see Figure 7c). The green color of the RDG isosurface compared to the blue color for RUVBIU discloses that the contact was weaker in PEPTAS, in line with the longer distance. The QTAIM/NCIplot analysis also revealed the existence of additional contacts due to the proximity of both counterions. The binding energy was very large (−225.8 kcal/mol) due to the ion-pair nature of the interaction. Orbital donor–acceptor interactions were also studied for this assembly using the NBO method. It showed the existence of a LP(Pt)→σ*(Sb–O) charge transfer with an almost negligible stabilization energy, $E^{(2)}$ = 0.14 kcal/mol (see Figure 7d). The NBO analysis also revealed the existence of a charge transfer from the LP at the Sb-atom to the antibonding σ*(Pt–N) orbital with a larger stabilization energy (1.28 kcal/mol). This suggests that the overall orbital charge transfer was from the Sb(III) to the Pt(II) atom, thus discarding the formation of a PnB bond in PEPTAS. To further demonstrate the nature of the Pt⋯Sb contact, the comparison of electron density (ED) and electrostatic potential (ESP) distribution along the Pt(II)⋯Sb(III) bond path was also performed. The plot shown in Figure 7e shows the positions of ED and ESP minima along the Pt(II)⋯Sb(III) bond path confirm the electron donation of Sb toward Pt(II) thus revealing the semi-coordination nature of the contact and that the Sb(III) was acting as the nucleophile via the LP.

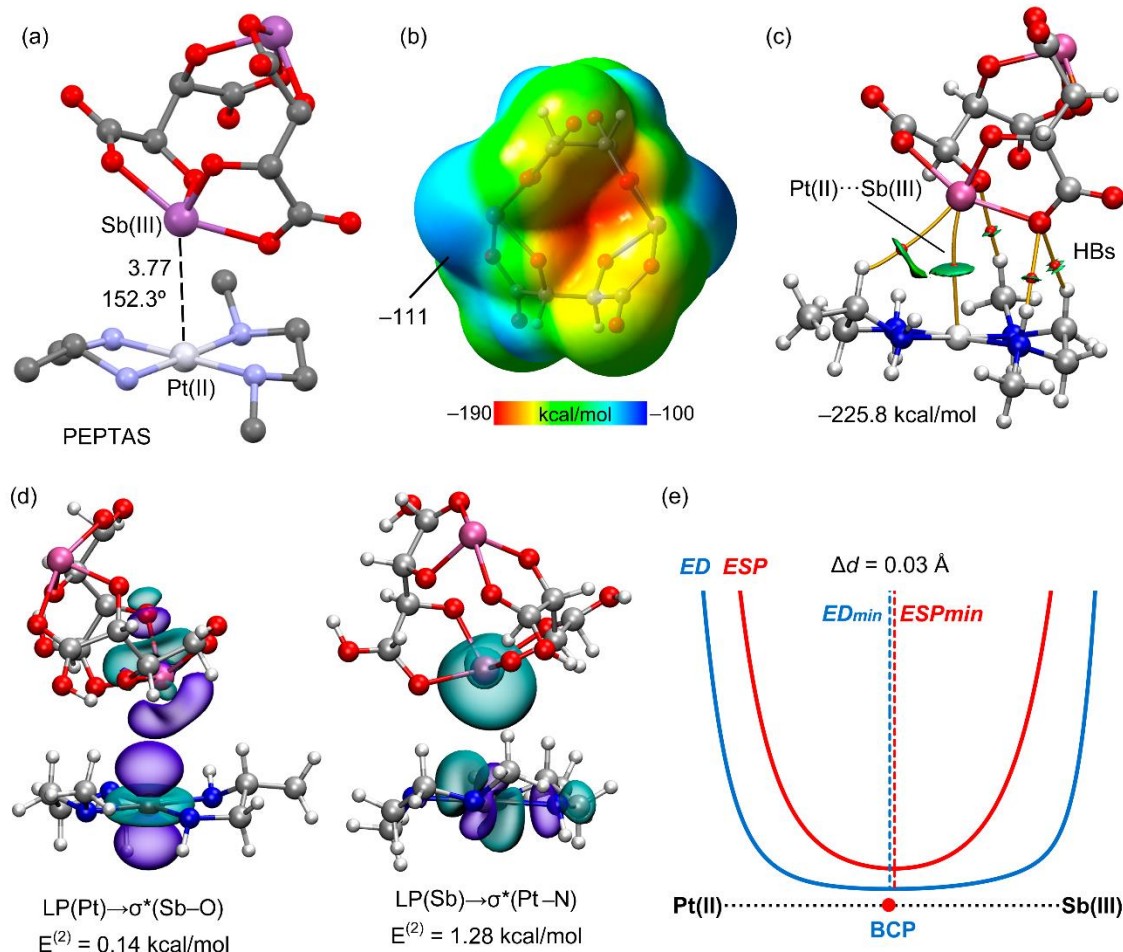

**Figure 7.** (**a**) X-ray structure of PEPTAS. Distances in Å. H-atoms omitted. (**b**) MEP surface of the $[SbCl_5]^{2-}$ unit. (**c**) QTAIM (bond CPs in red and bond paths as orange lines) and RDG isosurfaces of the salt. (**d**) NBO representations of the LP(Pt)→σ*(Sb–O) and LP(Sb)→σ*(Pt–N) interaction. (**e**) ED/ESP profile along the Pt(II)···Sb(III) bond path in PEPTAS.

## 3. Materials and Methods

In this manuscript, the Turbomole 7.0 program was used for the calculations [50]. The PBE0 [51]-D3 [52]/def2-TZVP [53,54] level of theory was selected for the calculations because it was used before to study similar interactions [20–25,55–57] including σ-hole halogen bonding involving metals. For Pt, the basis set def2-TZVP includes effective core potential for inner electrons [53,54]. To analyze the effect of relativistic effects, the interaction energy in RUVBIU was also estimated using the sarc-ZORA-TZVP [58] basis set Pt and TZP-ZORA [59] for Sb. The X-ray coordinates were used to evaluate the interactions in the solid state because we were interested in analyzing the interactions as they stood in the solid state, instead of finding the most favorable orientation between the molecules. The quantum theory of "atoms in molecules" (QTAIM) [60] and noncovalent interaction plot (NCIplot) [61] methods were used to characterize the interactions at the same level of theory. Both methods combined are very useful to reveal noncovalent interactions in real space and to disclose their attractive or repulsive nature. The following settings were used for the reduced density gradient (RDG) representations: $s = 0.40$, $-0.035$ a.u. $\leq$ (signλ$_2$)*ρ $\leq$ 0.035 a.u., density cut-off = 0.04 a.u. The natural bond orbital (NBO) analysis [62] was used to study charge transfer effects by means of the NBO7 program [63]. The molecular electrostatic potential (MEP) surfaces were plotted using the 0.001 a.u. isosurface. The QTAIM/NCIPlot/NBO analyses were plotted using the VMD software [64].

The investigation of the electron density (ED) and electrostatic potential (ESP) along the path that connects the Pt to the Sb was carried out using QTAIM methodology and the wave function files obtained from the Turbomole 7.0 package by means of the MultiWFN program [65]. The ED and ESP minima positions along the Pt···Sb path were determined and plotted by representing the data exported from the MultiWFN program [65] using the Excel program.

## 4. Conclusions

Several examples of X-ray structures exhibiting Pt···Halogen and Pt···Chalcogen were described in this manuscript. Moreover, this communication emphasizes the existence and relevance of PnBs involving five-coordinated Sb and Bi-atoms in different oxidation states (+3 and +5) in several X-ray structures retrieved from the Cambridge Structural Database (CSD). This manuscript also provides evidence of the ability of pentacoordinated Sb and Bi-atoms to establish highly directional and relevant PnBs in X-ray structures. More importantly, it also evidences for the first time the pnictogen bonding interactions in an X-ray structure where the Pt(II) metal center is acting as a nucleophile via the $d_z{}^2$ orbital. Moreover, it occurs in an ion-pair structure (RUVBIU) where the Sb belongs to the anionic specie. Therefore, the electron charge transfer that occurs from the $d_z{}^2$[Pt(II)] orbital to the antibonding σ*(Sb–Cl) orbital can be considered as a "charge unassisted" interaction where the anion is acting as a Lewis acid and the Pt(II) ion as a nucleophile. In addition to the geometrical evidence from the X-ray structure, additional support is obtained from DFT calculations, including ESD/ED comparison, MEP and QTAIM/NCIPlot computational tools.

**Supplementary Materials:** The following supporting information can be downloaded at: https://www.mdpi.com/article/10.3390/inorganics11020080/s1, Figure S1: MEP surface of the cationic part of RUVBIU. The points at selected parts of the surface are given in kcal/mol.

**Author Contributions:** Conceptualization, A.F.; methodology, A.B.; software, S.B., A.B., R.M.G. and A.F.; validation, R.M.G. and A.F.; investigation, S.B., A.B., R.M.G. and A.F.; resources, A.F.; writing—original draft preparation, A.F.; writing—review and editing, S.B., A.B., R.M.G. and A.F.; supervision, A.F.; project administration, A.F.; funding acquisition, A.F. All authors have read and agreed to the published version of the manuscript.

**Funding:** This research was funded by the "Ministerio de Ciencia, Investigacion y Universidades/Agencia Estatal de Investigación" (MICIU/AEI/10.13039/501100011033) of Spain (project PID2020-115637GB-I00. FEDER funds).

**Data Availability Statement:** Not applicable.

**Acknowledgments:** We thank the Centre de Tecnologies de la Informació (CTI) at University of the Balearic Islands (UIB) for the technical support.

**Conflicts of Interest:** The authors declare no conflict of interest.

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
