# Peer review of "Square Planar Pt(II) Ion as Electron Donor in Pnictogen Bonding Interactions"

_inorganics, doi:10.3390/inorganics11020080_

Round 1
Reviewer 1 Report
As was shown by the theoretical analysis of several structures by Hoffmann and Rogachev (JACS2013, 3262, which should be cited), the interaction of Pt(II) and I2 may occur in two ways, with diiodine serving as an electron acceptor (halogen-bond donor) in van Koten complexes and as an electron donor in the Cotton structure. Later, it was shown that d8 and d10 transition metals may act as nucleophiles in chalcogen bonding as well. The manuscript suggests the pnictogen bonding (PnB) with platinum (II) acting as PnB acceptor (nucleophile) in the CCDC structure RUVBIU. In comparison, Pt(II) serves as an electrophile in a very similar structure, PEPTAS. This very interesting conclusion was made based on the distinctions in the small E2 values obtained from NBO and relative locations of ED(min) and ESP(min).
The QTAIM analyses in Fig. 6 and 7 indicate the existence of the bonding, but they do not show which part is a nucleophile and which part is an electrophile. The authors have to provide more definitive evidence to support their conclusion that SbCl52- act as an electrophile and Pt(II) as a nucleophile.
First, the NBO analysis of RUVBIU shows a small contribution (1.07 kcal/mol) due to a donation from a LP orbital of Pt(II) located at the dz2 atomic orbital to the antibonding Sb–Cl orbital. However, NBO analysis frequently shows several contributions to E2 and a typical HaB complex can show contributions related to the charge transfer from the HaB donor to the acceptor and vice versa. If it is halogen bonding, the contributions related to the charge transfer to the MOs of the halogenated molecule are larger, and there is an overall charge transfer from the nucleophile to the electrophile. The authors should indicate if there were other contributions to E2 in their structures, and, if yes, list all of them in the SI. Additionally, they should show the direction and value of the overall charge transfer from one counter-part of the complex to another.
Second, the authors show that the relative locations of ESP and ED minima are different in the two structures. However, the separations of minima are very small, and they are shown qualitatively. The authors should provide a more detailed description of these characteristics, including their accuracy, to show that these small distinctions are meaningful.
Third, the authors showed MEP surface of SbCl52- in Figure 6. However, they did not show the corresponding MEP of Pt(II) complex. If there is an area of positive charge on the surface of Pt(II), it would be more logical to consider this center as a sigma-hole. In this way, it would fit definitions of the other sigma-hole interactions, with Pt(II) complex acting as an electrophile equivalent to HaB donor in halogen bonding. Accordingly, both complexes would represent analogs of coinage-metal bonding, with Pt(II) serving as an electrophilic center (the bond donor) and Sb(III) as a nucleophile (bond acceptor) with a small contribution of d[M]→σ*(X–Y) interactions in RUVBIU.
Author Response
First of all we would like to thank this referee for his/her careful reading of the manuscript, corrections and suggestions. Our responses are given below.
As was shown by the theoretical analysis of several structures by Hoffmann and Rogachev (JACS2013, 3262, which should be cited), the interaction of Pt(II) and I2 may occur in two ways, with diiodine serving as an electron acceptor (halogen-bond donor) in van Koten complexes and as an electron donor in the Cotton structure. Later, it was shown that d8 and d10 transition metals may act as nucleophiles in chalcogen bonding as well. The manuscript suggests the pnictogen bonding (PnB) with platinum (II) acting as PnB acceptor (nucleophile) in the CCDC structure RUVBIU. In comparison, Pt(II) serves as an electrophile in a very similar structure, PEPTAS. This very interesting conclusion was made based on the distinctions in the small E2 values obtained from NBO and relative locations of ED(min) and ESP(min).
Reply: Thank you very much for the comments and the reference. It has been now cited in the introduction (ref 26).
The QTAIM analyses in Fig. 6 and 7 indicate the existence of the bonding, but they do not show which part is a nucleophile and which part is an electrophile. The authors have to provide more definitive evidence to support their conclusion that SbCl52- act as an electrophile and Pt(II) as a nucleophile.
Reply: We have done this using the relative locations of ED(min) and ESP(min) along the path connecting the Pt and Sb atoms. We are unable to figure out how to provide additional support. In case the referee has some additional suggestion how to prove or disprove this conclusion, we will be happy to do it. It can be also left as an open question for the scientific community to investigate. In any case, please note the X-ray structures of HEKNUC and RAHFEN in Figure 5 show a typical lone pair donor atom (oxygen of amide or ether) that is located opposite to the Sb–Cl,I bond similar to that it is found for the Pt(II) in RUVBIU, thus suggesting that the Pt(II) is playing the same role.
First, the NBO analysis of RUVBIU shows a small contribution (1.07 kcal/mol) due to a donation from a LP orbital of Pt(II) located at the dz2 atomic orbital to the antibonding Sb–Cl orbital. However, NBO analysis frequently shows several contributions to E2 and a typical HaB complex can show contributions related to the charge transfer from the HaB donor to the acceptor and vice versa. If it is halogen bonding, the contributions related to the charge transfer to the MOs of the halogenated molecule are larger, and there is an overall charge transfer from the nucleophile to the electrophile. The authors should indicate if there were other contributions to E2 in their structures, and, if yes, list all of them in the SI. Additionally, they should show the direction and value of the overall charge transfer from one counter-part of the complex to another.
Reply: Since the supramolecular dimer is formed between a dication and a dianion, the overall charge transfer goes from the cation to the anion as expected. In fact, in the complex the charge of the cation is +1.7 e and that of the anion is -1.7 e (0.3 e transfer from the anion to the cation, as expected. What we want to highlight in this manuscript is that the particular Pt···Sb contact goes in the opposite direction. There is not any donation from the LP(Sb) to any non rydberg Pt-orbitals in the NBO analysis. There is an additional contribution from the σ(Pt–N) orbitals to the σ*(Sb–Cl) orbital ( 1.15 kcal/mol) that has been commented in the revised main text. All contributions are now listed in the ESI.
Second, the authors show that the relative locations of ESP and ED minima are different in the two structures. However, the separations of minima are very small, and they are shown qualitatively. The authors should provide a more detailed description of these characteristics, including their accuracy, to show that these small distinctions are meaningful.
Reply: We have provided in the figures the Δd values that are indeed very small, 0.05 and 0.03 Å, for RUVBIU and PEPTAS respectively. Regarding the accuracy, a much deeper theoretical analysis would be needed to investigate the accuracy, that is obviously out of the scope of the present investigation. This methodology and the Δd values have used recently to estimate interaction energies (10.3390/molecules27154848) of phosphine oxides as electron donors. Moreover, the results are in good agreement with the conclusions derived from the well established NBO method.
Third, the authors showed MEP surface of SbCl52- in Figure 6. However, they did not show the corresponding MEP of Pt(II) complex. If there is an area of positive charge on the surface of Pt(II), it would be more logical to consider this center as a sigma-hole. In this way, it would fit definitions of the other sigma-hole interactions, with Pt(II) complex acting as an electrophile equivalent to HaB donor in halogen bonding. Accordingly, both complexes would represent analogs of coinage-metal bonding, with Pt(II) serving as an electrophilic center (the bond donor) and Sb(III) as a nucleophile (bond acceptor) with a small contribution of d[M]→σ*(X–Y) interactions in RUVBIU.
Reply: Thank you for this comment. The interaction in RUVBIU cannot be defined as a coinage bond because it is the σ-hole of the SbCl52- molecule that is pointing to the Pt instead of a LP of any of the Cl-atoms. This fact suggests that this contact is not a coinage bond. Moreover, the MEP plot of the Pt complex does not show a σ-hole at the Pt-atom. Indeed the Pt is the second most nucleophilic part of the molecule (the global MEP minimum is located at the O-atoms of the nitro group). The MEP has been provided in the ESI (Figure S1).
Reviewer 2 Report
It is very solid computational work which highlightsthe importance of pnictogen bonding in transition metal systems. It would be worthy to comment before the acceptance:
1. What is the nature of Sb···P contact in terms of
London dispersion contribution (clearly Lp-pi
interactions are present) as well as electrostatic
terms.
2. How strongly relativistic effects contribute to
Sb···P contacts.
Overall very nice piece of work which deserves to be published in Inorganics.
Author Response
We thank this referee for his/her careful reading of the manuscript, corrections and suggestions. We have revised the mansuscript accordingly, as detailed below:
It is very solid computational work which highlights the importance of pnictogen bonding in transition metal systems. It would be worthy to comment before the acceptance:
- What is the nature of Sb···P contact in terms ofLondon dispersion contribution (clearly Lp-pi interactions are present) as well as electrostatic terms.
Reply: We have estimated the contribution of dispersion, which is quite large due to the presence of the lp-pi interactions as highlighted by the referee. We have included the new results in Figure 6 and commented in the text
- How strongly relativistic effects contribute toSb···P contacts.
Reply: Relativistic effects are also important for the interaction energy, as revealed by the new calculations, see Figure 6 and the main text
Overall very nice piece of work which deserves to be published in Inorganics.
Reply: Thank you for supporting publication
Reviewer 3 Report
The paper by Frontera et al. represents the computation investigation of the non-covalent interactions on the example of known Pt(II) complexes with pnictogen atoms. The paper is well-written and the results support the conclusions. I have read the article with great interest. I can't find any mistakes or give any notes. The article is suitable for publication.
One small suggestion:
Please, give a color-codes or names for the coordinated atoms in all figures.
Author Response
We would like to thank this reviewer for his/her careful reading of the manuscript, and suggestion. The change made is detailed below
The paper by Frontera et al. represents the computation investigation of the non-covalent interactions on the example of known Pt(II) complexes with pnictogen atoms. The paper is well-written and the results support the conclusions. I have read the article with great interest. I can't find any mistakes or give any notes. The article is suitable for publication.
One small suggestion:
Please, give a color-codes or names for the coordinated atoms in all figures.
Reply: Done
Round 2
Reviewer 1 Report
The explanations provided by the authors are satisfactory and the article can be published.